# Prevalence, risk and protective factors of burnout among Korean hospitalists

**Kyung Mee Park**[1◎], **Jaewoong Kim**[2◎], **Taeyoung Kyong** ●[1], **Hee Youn Han**[1],
**Song Yi Song** ●[1], **Se Yoon Park** ●[3]*

**1** Department of Hospital Medicine, Yongin Severance Hospital, Yonsei University College of Medicine, Yongin, Korea, **2** Department of Biomedical Systems Informatics, Yonsei University College of Medicine, Seoul, Korea, **3** Department of Internal Medicine, Hanyang University College of Medicine, Seoul, Korea

◎ These authors contributed equally to this work.
* livinwill2@gmail.com

## Abstract

### Background

Burnout among healthcare professionals is a critical factor which affects patient safety, treatment outcomes, and the quality of care. This is especially important for hospitalists who manage inpatient care, yet no studies have been conducted on this issue in Korea. This study aimed to investigate burnout and psychiatric symptoms among hospitalists in Korea, as well as to identify the risk and protective factors associated with these issues.

### Materials and methods

A cross-sectional online survey was conducted targeting all hospitalists in Korea (n=303), and 24.1% (n=79) completed the survey. The Maslach Burnout Inventory–Human Services Survey was used to measure burnout; the Depression, Anxiety, and Stress Scales was used to assess psychiatric symptoms; and the Insomnia Severity Index was used to evaluate sleep disturbances. Risk and protective factors against burnout were assessed using a 5-point Likert scale.

### Results

More than half of the respondents reported high graded burnout for two domains: depersonalization (50.6%) and reduced personal accomplishment (57%). Conflicts with caregivers, excessive workload, and long working hours were common risk factors for both burnout domains. The satisfaction with nonclinical work was identified as protective factor in depersonalization, and the availability of a research mentor and cap on daily inpatient load per hospitalist were protective factors in reduced personal accomplishment. In the correlation analysis, the maximum number of inpatients and hospitalists per hospital was a significant factor in reducing burnout.

**Data availability statement:** Due to the potential risk of identifying individual participants and the sensitive nature of the collected information, the data from this study cannot be publicly shared. However, the full survey questionnaire is available as supplemental information. Researchers who wish to access the data may submit individual requests to the corresponding author (livinwill2@gmail.com) or the Institutional Review Board of Yongin Severance Hospital (ysirb@yuhs.ac). Data sharing will be considered in accordance with ethical guidelines and institutional policies.

**Funding:** This study was supported by the Korea Medical Device Development fund under grant number (RS-2023-00255005) and by intramural research funding for general professor in Yonsei University College of Medicine (6-2021-0224). Prof. Kyung Mee Park received funding from both sources. The funders had no role in study design, data collection and analysis, decision to publish, or preparation of the manuscript.

**Competing interests:** The authors have declared that no competing interests exist.

## Conclusions

This study revealed a high graded burnout rate of more than 50% in depersonalization and reduced personal accomplishments domain among Korean hospitalists, and found the risk and protective factors against burnout. The development of targeted interventions to mitigate burnout based on this study could enhance the mental well-being of healthcare professionals and improve the overall quality of medical care.

A cross-sectional online survey was conducted targeting all hospitalists in Korea (n=303), and 24.1% (n=79) completed the survey. The Maslach Burnout Inventory–Human Services Survey was used to measure burnout; the Depression, Anxiety, and Stress Scales was used to assess psychiatric symptoms; and the Insomnia Severity Index was used to evaluate sleep disturbances. Risk and protective factors against burnout were assessed using a 5-point Likert scale.

More than half of the respondents reported high graded burnout for two domains: depersonalization (50.6%) and reduced personal accomplishment (57%). Conflicts with caregivers, excessive workload, and long working hours were common risk factors for both burnout domains. The satisfaction with nonclinical work was identified as protective factor in depersonalization, and the availability of a research mentor and cap on daily inpatient load per hospitalist were protective factors in reduced personal accomplishment. In the correlation analysis, the maximum number of inpatients and hospitalists per hospital was a significant factor in reducing burnout. This study revealed a high graded burnout rate of more than 50% in depersonalization and reduced personal accomplishments domain among Korean hospitalists, and found the risk and protective factors against burnout. The development of targeted interventions to mitigate burnout based on this study could enhance the mental well-being of healthcare professionals and improve the overall quality of medical care.

## Introduction

There has been a growing number of hospitalists worldwide, supported by evidence suggesting that hiring more hospitalists can result in cost effectiveness, higher quality of care, and superior treatment outcomes.[1–3] According to Howell et al., the number of hospitalists in the United States increased by more than 50% between 2012 and 2019, resulting in 44,037 hospitalists.[4] In Korea, a pilot version of the hospitalist system was initiated in 2016, and the official Korean hospitalist system was implemented in 2020. Consequently, the number of hospitalists increased from 5 in 2016–249 by 2020.[5] The number of hospitalists continues to grow, with 303 on duty in Korea as of February 2023.[6]

Despite the benefits of introducing hospitalists to hospitals, focusing solely on their productivity can potentially cause stress and eventual burnout. Burnout is a

state of psychological, emotional, and physical exhaustion caused by persistent and excessive work-related stress and pressure.[7,8] Medical professionals are vulnerable to burnout. Although interpretation requires caution due to differences in assessment methods, measurement tools, and study populations [9], according to a review of burnout among medical residents, the burnout rate varied from 17.6% to 76.0%.[10] Another study claimed that burnout among doctors ranged from 12% to 61%.[11] Hospitalists also experience burnout symptoms. According to the Hospitalist Worklife Survey by Wetterneck et al., 30% of 794 hospitalists reported burnout symptoms,[12] and Glisch et al. found that 62% of 52 academic hospitalists experienced burnout.[13] Another study claimed that the burnout rate among internal medicine hospitalists was 52.3%, which showed no significant difference when compared with outpatient internalists.[14]

Healthcare professionals with burnout may experience a decrease in decision-making abilities, difficulty in maintaining relationships with patients or co-workers, and an increase in medical errors.[10,15–17] Additionally, prolonged burnout can lead to deterioration of mental health, which can manifest as other psychiatric symptoms such as depression, anxiety, and insomnia.[18–21] Recognizing and addressing psychiatric symptoms, in addition to the possibility of burnout among healthcare professionals, is not merely an ethical concern but also a strategic investment in the overall effectiveness of the healthcare system. Considering that the primary objective of hiring hospitalists is to improve the quality of healthcare, it is vital to ensure that they experience good mental health.

As the clinical competence of hospitalists directly affects the quality of care for hospitalized patients, burnout and mental health issues can be deemed even more critical. Despite the importance of understanding the impact of burnout and mental health on physicians, no study has specifically targeted Korean hospitalists. Given the expected continuous increase in the number of hospitalists in Korea, there is an urgent need to address the mental well-being of hospitalists and establish an appropriate healthcare environment. In this study, we aimed to investigate the burnout, and psychiatric symptoms of hospitalists using an organized questionnaire. Additionally, we sought to identify the risk and protective factors associated with these issues.

## Materials and methods

### Study design and target population

The study was a cross-sectional survey conducted over 20 days, from January 30 to February 18, 2023. Among 384 registered Korean hospitalists as of February 2023, the target population comprised 303 hospitalists who were members of the Korean Association of Internal Medicine or the Korean Society of Hospital Medicine. The online survey link was sent to the participants through e-mail and text messages by the Korean Association of Internal Medicine and Korean Society of Hospital Medicine. To encourage participation, reminders were sent on the fourth, tenth, and fourteenth days. All respondents were anonymized, and their answers were confidential. Completion of the survey was voluntary, and only one response from each participant was accepted. This survey research was conducted following the Checklist for Reporting of Survey Studies (CROSS) guidelines[22]. Written informed consent was obtained from all the participants online. This study was approved by the Institutional Review Board (IRB) of Yongin Severance Hospital (9-2022-0176).

### Measurement items

Measured items included demographic information, such as sex and age, survey information related to the career of a hospitalist, basic information about the working hospital, and clinical information, such as new admissions and average daily inpatients. Potential risk and protective factors against burnout and psychiatric symptoms were gathered through survey items using a 5-point Likert scale. The questionnaires were divided into three categories: work- and life-related satisfaction, family and social support, and coping skills and abilities. Details of the questionnaire used in this study are presented in S1 File.

Burnout was measured using 22 items from the previously validated Korean version of the Maslach Burnout Inventory–Human Service Survey (MBI-HSS).[23] Each subdimension was divided into emotional exhaustion, depersonalization, and reduced personal accomplishment, and each category consisted of seven, seven, or eight items, respectively. A 7-point Likert scale, where 1 represents "not at all" and 7 represents "every day," was used to measure burnout items, with higher scores indicating a higher degree of burnout, except for reduced personal accomplishment. The personal accomplishment section comprised items with reverse questions. Consequently, higher scores indicated lower burnout. Burnout severity was categorized as low, moderate, or high. For emotional exhaustion items, a score of 0–18 is classified as "Low," 19–26 as "Moderate," and 27 or above as "High." Depersonalization items are classified as "Low" for scores of 0–5, "Moderate" for scores of 6–9, and "High" for scores of 10 or above. For personal accomplishment items, scores of 40 or above are classified as "Low," 34–39 as "Moderate," and 0–33 as "High."

The Depression, Anxiety, and Stress Scales (DASS-21)[24] was used to measure psychiatric symptoms. The DASS scale uses a 4-point Likert scale, where 0 means "did not apply at all" and 3 means "applied most of the time." Higher scores indicated higher levels of depression, anxiety, and stress. Each subscale (Depression, Anxiety, and Stress) consists of seven items. Depression severity is defined as follows: 0–9 points as "Normal," 10–13 as "Mild," 14–20 as "Moderate," 21–27 as "Severe," and 28 or above as "Extremely Severe." Anxiety is categorized as "Normal" (0–7), "Mild" (8–9), "Moderate" (10–14), "Severe" (15–19), and "Extremely Severe" (20 or above). Stress severity is classified as "Normal" (0–14), "Mild" (15–18), "Moderate" (19–25), "Severe" (26–33), and "Extremely Severe" (34 or above).

The Insomnia Severity Index (ISI)[25] was used to assess sleep quality. The ISI consists of seven items rated on a 5-point Likert scale. Items 1–3 are rated from "none" to "very severe," item 4 is rated from "very satisfied" to "very dissatisfied," and items 5–7 are rated from "not at all" to "very much." The total ISI score is calculated as the sum of the scores for all seven items, with higher scores indicating a more severe level of insomnia. Clinical significance was categorized as no insomnia (0–7), subthreshold insomnia (8–14), and moderate to severe insomnia (15–28).

Survey item Cronbach's alpha was calculated to measure the internal reliability of the burnout, DASS-21, and ISI scales. The results confirmed that all had values above 0.6, indicating high reliability.

### Statistical analysis

Descriptive statistical analysis for categorical variables was presented using relative frequencies, and continuous variables were presented using medians and interquartile ranges (1st and 3rd quartiles). Based on the normality assessment of each independent variable, parametric methods (independent t-test, ANOVA) or non-parametric tests (Mann-Whitney U test, Kruskal-Wallis test) were employed. Univariate analysis was conducted on total burnout score as well as on emotional exhaustion, depersonalization, and reduced personal accomplishment, and only variables with a significance level of less than 0.05 were utilized as predictors in the multiple linear regression analysis. Multiple regression analyses were performed individually for total burnout score, emotional exhaustion, depersonalization, and reduced personal accomplishment. For categorical variables, dummy coding was applied prior to analysis. Statistical analyses were performed using IBM Statistical Package for Social Sciences (SPSS) version 28.0 (SPSS for Window version; SPSS Inc., Chicago, IL, USA) and R version 4.2.2 (http://www.R-project.org). All statistical tests were two-sided, and a p-value of < 0.05 was considered statistically significant.

## Results

### Participant characteristics

Among the 303 hospitalists, 79 completed the survey, resulting in a response rate of 24.1%. General information and work-related characteristics of the respondents is presented in Table 1 and S1 Table. The median age of the respondents was 39 years, and 51.9% (n=41) were male. The most common trainee department was the internal medicine department (54.4%,

**Table 1. Demographic features of Korean hospitalists.**

| Characteristics | Respondents (n=79) |
|---|---|
| Age, median (IQR) | 39.5 (36–45) |
| Sex, male | 41 (51.9) |
| Married | 58 (73.4) |
| Having children | 52 (65.8) |
| Trainee department | |
| Internal medicine | 43 (54.4) |
| Surgery | 14 (17.7) |
| Pediatrics | 8 (10.1) |
| Family medicine | 6 (7.6) |
| Obstetrics and gynecology | 2 (2.6) |
| Others | 6 (7.6) |
| Fellowship training | 54 (68.4) |
| Career as a hospitalist | |
| Less than 1 year | 10 (12.7) |
| Over 1 year, under 2 years | 16 (20.3) |
| Over 2 years, under 3 years | 14 (17.7) |
| Over 3 years, under 4 years | 22 (27.8) |
| Over 4 years, under 2 years | 5 (6.3) |
| Over 5 years | 12 (15.2) |
| Position | |
| Professor | 4 (5.1) |
| Associate professor | 10 (12.7) |
| Assistant professor | 19 (24.1) |
| Others (clinical professor, etc.) | 46 (58.1) |
| Classification of hospitals | |
| Tertiary | 55 (69.6) |
| General | 24 (30.4) |
| Size of hospitals | |
| Less than 900 beds | 38 (48.1) |
| 900–1200 beds | 13 (16.5) |
| 1200 beds or more | 28 (35.4) |
| Location | |
| Seoul and Gyeonggi Province | 39 (88.6) |
| Others | 9 (11.4) |
| Classification of affiliation | |
| General internal medicine or surgery | 35 (44.3) |
| Department of hospital medicine | 28 (35.4) |
| Individual department | 16 (20.3) |

Data are expressed as frequency (%), unless otherwise indicated.

Abbreviation: IQR, interquartile range.

n=43), and approximately two-thirds of the respondents had finished fellowship training (68.4%, n=54). The most common career length as a hospitalist was over three years and under four years (27.8%, n=22). Most respondents worked in the capital areas of Korea (Seoul and Gyeonggi Provinces, 88.6%, n=70). The respondents reported an average daily inpatient

caseload of 15, with average weekly working hours of 45. The cap on daily inpatient load per hospitalist ranged from 15 to 23, with a median of 19. Approximately 55.7% of respondents (n=44) worked alongside physician assistant nurses, and 19.0% (n=15) worked alongside residents. Only 17.7% of respondents (n=14) reported having a research mentor.

### Burnout and psychiatric symptoms among hospitalists

More than half of the respondents exhibited high-severity scores in two burnout domains: 50.6% (n=40) for depersonalization, 57.0% (n=45) for reduced personal accomplishment, and 10.1% (n=8) reported high-severity scores of emotional exhaustion (Fig 1). Of the respondents, 7.5% (n=6) reported mild to moderate depressive symptoms and 5.1% (n=4) reported mild to moderate anxiety symptoms. None of the respondents screened positive for severe depression, anxiety, or stress. Approximately 30.4% (n=24) of the respondents reported experiencing subthreshold insomnia and 8.9% (n=7) were screened for moderate to severe insomnia (Table 2). The correlation analysis between each burnout component, DASS, and ISI demonstrated statistically significant associations, except for the correlation between Reduced Personal Accomplishment and ISI. The absolute values of the correlation coefficients between the components of Burnout and DASS-Depression were all below 0.6, indicating no concerns regarding discriminant validity (S2 Table).[26]

### Protective and risk factors related to burnout

A cap on daily inpatient load per hospitalist ($\beta = -0.477$, $p = 0.032$, availability of research mentor ($\beta = -12.427$, $p = 0.034$, and satisfaction with non-clinical work ($\beta = -9.347$, $p = 0.062$ were found to be protective factors for burnout in multivariable linear regression analysis. The cap on daily inpatient load per hospitalist and the availability of research mentors were significantly related to low total burnout scores and reduced personal accomplishment, while satisfaction with non-clinical work was significantly related to low total burnout scores and low depersonalization scores (Table 3).

In the correlation analysis, the cap on daily inpatient load per hospitalist and hospitalists per hospital was a significant factor in reducing burnout. The cap on daily inpatient load per hospitalist was significantly associated with low total burnout scores ($p = 0.023$) and personal accomplishment ($p = 0.042$), whereas the number of hospitalists per hospital was significantly associated with personal accomplishment ($p = 0.040$). The number of hospitalists per hospital also tended to be associated with low total burnout scores ($p = 0.105$, S1 Fig).

Factors contributing to dissatisfaction with work, as reported by respondents experiencing high emotional exhaustion, included the severity of patients (n=7, 88%), long working hours (n=5, 62.5%), and low salary (n=6, 75.0%). In the high depersonalization group, conflict with caregivers (n=19, 47.5%), excessive workload (n=17, 42.5%), and long working hours (n=17, 42.5%) were the most common sources of dissatisfaction. In the group with highly reduced personal accomplishment scores, the severity of patient and guardian responses accounted for the highest proportion, followed by excessive workload and long working hours. (S3 Table).

## Discussion

Approximately 50% of respondents exhibited signs of burnout, particularly in the domains of depersonalization and personal accomplishment. This finding suggests a concerning prevalence of burnout among Korean hospitalists and highlights the need for targeted interventions and support to address the challenges faced in their demanding roles. The maximum number of hospitalists, number of hospitalists per hospital, cap on daily inpatient load per hospitalist, availability of research mentors, satisfaction with nonclinical work, and number of hospitalists per hospital were found to protect against burnout. Conversely, conflicts with caregivers, excessive workload, and long working hours were identified as risk factors for both depersonalization and personal accomplishment. Additionally, approximately 30% of specialists exhibited subthreshold insomnia, while less than 10% of surveyed hospitalists reported severe insomnia. Clinically significant levels of depression and anxiety were rarely observed.

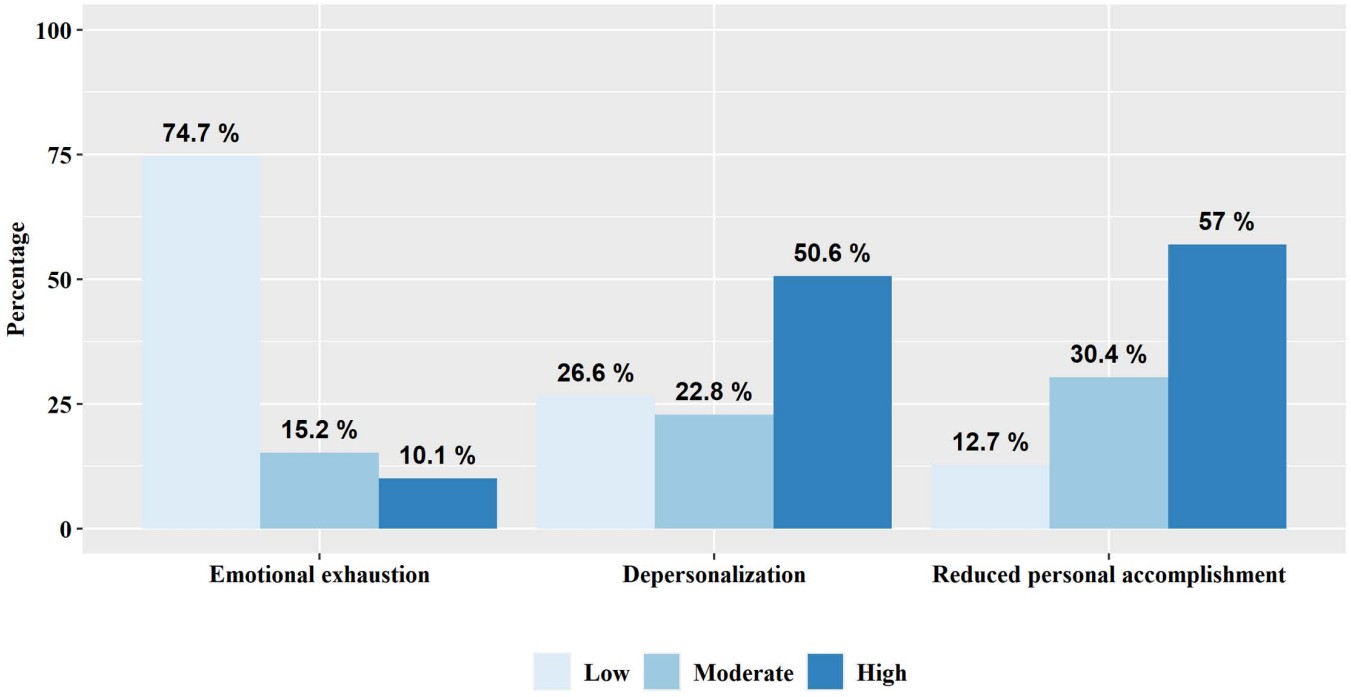

**Fig 1. Burnout rate of Korean hospitalists (n=79).**

**Table 2. Burnout and psychiatric symptoms of Korean hospitalists.**

| Characteristics | n (%) | | |
|---|---|---|---|
| Burnout | Emotional exhaustion | Depersonaliza-tion | Reduced personal accomplishment |
| High | 8 (10.1) | 40 (50.6) | 45 (57.0) |
| Moderate | 12 (15.2) | 18 (22.8) | 24 (30.4) |
| Low | 59 (74.7) | 21 (26.6) | 10 (12.7) |
| DASS | Depres-sion | Anxiety | Stress |
| Normal | 73 (92.4) | 74 (94.9) | 78 (100.0) |
| Mild | 5 (6.3) | 3 (3.8) | 0 |
| Moderate | 1 (1.3) | 1 (1.3) | 0 |
| Severe | 0 | 0 | 0 |
| Extremely Severe | 0 | 0 | 0 |
| ISI | | Insomnia symp-tom severity | |
| No insomnia | 46 (59.7) | | |
| Sub-threshold insomnia | 24 (30.4) | | |
| Moderate to severe insomnia | 7 (8.9) | | |

Abbreviations: DASS, Depression, Anxiety, and Stress Scales; ISI, Insomnia Severity Index.

**Table 3. Multivariable linear regression with burnout total score and subcategory score.**

| Characteristics | Total burnout score | | | Emotional exhaustion | | | Depersonalization | | | Reduced personal accomplishment | | |
|---|---|---|---|---|---|---|---|---|---|---|---|---|
| | β (SE) | Bª | P value | β (SE) | B | P value | β (SE) | B | P value | β (SE) | B | P value |
| Cap on daily inpatient load per hospitalist | -0.477 (0.22) | -0.267 | 0.032 | -0.127 (0.11) | -0.148 | 0.235 | -0.125 (0.07) | -0.227 | 0.068 | 0.225 (0.10) | 0.290 | 0.024 |
| Involvement of physician assistant nurses | | | | | | | | | | | | |
| Yes | -4.685 (5.16) | -0.109 | 0.368 | -4.513 (2.53) | -0.220 | 0.080 | -2.569 (1.60) | -0.195 | 0.114 | -2.397 (2.31) | -0.129 | 0.304 |
| No | Ref. | | | Ref. | | | Ref. | | | Ref. | | |
| Availability of research mentor | | | | | | | | | | | | |
| Yes | -12.427 (5.72) | -0.260 | 0.034 | -4.373 (2.80) | -0.191 | 0.124 | -2.506 (1.77) | -0.171 | 0.164 | 5.549 (2.56) | 0.268 | 0.035 |
| No | Ref. | | | Ref. | | | Ref. | | | Ref. | | |
| Satisfaction with non-clinical work | | | | | | | | | | | | |
| Satisfied | -9.347 (4.90) | -0.231 | 0.062 | -4.682 (2.40) | -0.242 | 0.056 | -3.177 (1.52) | -0.255 | 0.041 | 1.488 (2.20) | 0.085 | 0.501 |
| Non-satisfied | Ref. | | | Ref. | | | Ref. | | | Ref. | | |
| F-statistic Adjusted $R^2$ Durbin-Watson | 4.082 0.173 1.911 | | <0.001 | 3.290 0.134 2.093 | | 0.017 | 3.765 0.158 2.000 | | <0.001 | 2.895 0.174 1.885 | | 0.030 |

Abbreviation: SE, standard error.

ªStandardized β coefficient.

The signs of burnout among hospitalists in our study, which could be lead to higher risk for burnout in Korean hospitalists, closely aligns with that found in previous research (30% to 60%),[12–14] including the findings in Roberts et al. (52.3% among 130 internal-medicine hospitalists).[14] The difference from previous studies lies in the fact that, while they focused on internal medicine hospitalists, we targeted hospitalists with diverse specialties. Despite this distinction, the similarity in the findings suggests a substantial burnout rate among hospitalists, irrespective of their specialization. In a previous study, high levels of emotional exhaustion and depersonalization were reported,[14] whereas in the present study, elevated levels of depersonalization and reduced personal accomplishment with relatively low levels of emotional exhaustion were observed. Depersonalization and personal accomplishment are more associated with resources, including social support or the quality of job contents, while emotional exhaustion tends to be strongly associated with job demand variables, including work overload.[27] Thus, the result of this study indicates that burnout among Korean hospitalists may be influenced more by the quality of their work than by the quantity of their workload.

In terms of protective factors, our findings confirmed that a higher number of hospitalists per hospital contributes to reducing burnout. This stability is facilitated by the presence of colleagues who can perform similar tasks, forming a consensus within the hospital, and reducing the overall workload. Satisfaction with nonclinical work and the presence of research mentors significantly contributed to reducing burnout. This suggests that when hospitalists focus exclusively on managing inpatients without experiencing satisfaction from other tasks, there could be an increased risk of burnout. To establish a sustainable work environment for hospitalists, it is imperative to enhance their satisfaction with various tasks beyond inpatient care by providing them with adequate support. These findings align with the results of previous studies, indicating that high job satisfaction, strong leadership or mentorship, and appropriate peer support could act as protective

factors against hospitalist burnout.[12,13] Additionally, a higher cap on daily inpatient load per hospitalist was correlated with reduced burnout, specifically contributing to an increase in personal accomplishment within burnout domains. Considering the relatively short history of the domestic hospitalist system, it can be interpreted that achieving personal accommodation is facilitated by direct patient care, autonomous treatment, and appropriate workload allocation. However, it is worth noting that contrary to the general expectation that higher workload, which might relate to higher patient volumes, could worsen burnout. Further research is needed to explore this discrepancy.

In this study, conflicts with caregivers, excessive workload, and long working hours were common risk factors for both depersonalization and reduced personal accomplishment. Studies have suggested that factors including prolonged working hours, dissatisfaction with work, home-work interaction conflict, high stress levels, emotional interactions with patients, and the cognitive demands of duties are suspected risk factors for burnout in doctors, which is consistent with our study results.[11,28] Certain demographic factors, including young age, female sex, and negative marital status are also risk factors for vulnerability to burnout.[28] However, in this study, no significant correlations were observed between demographic factors and burnout. Considering the job characteristics of hospitalists primarily responsible for patient- and family centered care, it is evident that conflicts with patient caregivers could serve as a primary factor contributing to burnout.[29]

No significant increase was found in depression, anxiety, or stress symptoms among Korean hospitalists. This is a positive outcome, suggesting that the current manifestations of burnout symptoms are not chronic enough to induce psychiatric symptoms. However, it is noteworthy that approximately 30% of respondents experienced subthreshold insomnia, a rate higher than that observed in the general population, which is approximately 10%.[30] Vela-Bueno et al. reported a clear association between elevated burnout levels and impaired sleep quality among physicians.[31] Consequently, it is plausible that the insomnia observed in Korean hospitalists is linked to burnout, underscoring the need to address and reduce burnout within this professional cohort.

There are some limitations in this study. First, the design of this study is cross-sectional, which makes it difficult to determine temporal sequences between variables. Second, the overall response rate of 24.1% cannot entirely rule out the possibility of selection bias, which may affect the generalizability of the results to the entire population of Korean hospitalists. Considering these aspects, it may be challenging to assert that this study comprehensively captured the situation of burnout of all Korean hospitalists.

Nonetheless, the significance of this study stems from its distinction as the first to examine burnout and psychiatric symptoms among hospitalists during the early adoption phase of the hospitalist system in Korea. Unlike the United States, where the system has been established for over 20 years and is relatively well-defined, this study explores the experience of South Korea, where the hospitalist system is still in its early stages of implementation. The findings of this study hold significance as they may serve as a valuable reference for other countries that are considering implementing the hospitalist system or are in the initial phase of its adoption.

In conclusion, this study uncovers the significant issue of burnout and its protecting factors among Korean hospitalists. This highlights the urgent need for strategies to address mental health concerns in this professional group. The findings can guide the development of targeted interventions to mitigate burnout, ultimately enhancing the mental well-being of healthcare professionals and improving the overall quality of medical care.

## Supporting information

**S1 File. Survey Questionnaire.**
(DOCX)

**S1 Table. Work-related characteristics of Korean hospitalists.**
(DOCX)

**S2 Table. Correlation analysis between burnout components, DASS, and ISI.**
(DOCX)

**S3 Table. Subjective reported factors related to burnout of the Korean hospitalists.**
(DOCX)

**S1 Fig. Correlation between the burnout symptoms and the number of hospitalists.**
(DOCX)

## Acknowledgments

We sincerely thank the Korean hospitalists for dedicating their time to make this project possible. The authors are grateful to the Korean Society of Hospital Medicine and the Korean Society of Surgical Hospital Medicine for identifying target participants for the survey.

## Author contributions

**Conceptualization:** Kyung Mee Park, Se Yoon Park.

**Data curation:** Kyung Mee Park, Jaewoong Kim, Hee Youn Han, Song Yi Song, Se Yoon Park.

**Formal analysis:** Jaewoong Kim.

**Investigation:** Kyung Mee Park, Hee Youn Han, Song Yi Song.

**Methodology:** Kyung Mee Park, Jaewoong Kim, Taeyoung Kyong.

**Project administration:** Se Yoon Park.

**Software:** Jaewoong Kim.

**Validation:** Taeyoung Kyong, Se Yoon Park.

**Writing – original draft:** Kyung Mee Park, Jaewoong Kim.

**Writing – review & editing:** Kyung Mee Park, Jaewoong Kim, Taeyoung Kyong, Hee Youn Han, Song Yi Song, Se Yoon Park.

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
