## [Decision Letter · Decision Letter 0]

29 Oct 2024

PONE-D-24-45133Prevalence, Risk and Protective Factors of Burnout among Korean HospitalistsPLOS ONE

Dear Dr. Park,

Thank you for submitting your manuscript to PLOS ONE. After careful consideration, we feel that it has merit but does not fully meet PLOS ONE’s publication criteria as it currently stands. Therefore, we invite you to submit a revised version of the manuscript that addresses the points raised during the review process.

Please answer to all comments by reviewer 2.  Other comments: - In Table 1, it is unclear what the p-values refer to, as no comparisons are explicitly mentioned. Additionally, it would be helpful to compare the characteristics of respondents with those of the full sample of 303 hospitalists to whom the questionnaire was sent. This comparison could provide insight into any potential response bias and the representativeness of the study sample.- The limitations section should be expanded to provide a more comprehensive discussion of the study's constraints. Specifically, it should be noted that the cross-sectional design limits any causal inferences, as it only allows for an association between variables to be observed at a single point in time. Additionally, the response rate of 24.1% introduces the possibility of selection bias, which could affect the generalizability of the results to the entire population of hospitalists in Korea. Including these points would help clarify the scope and limitations of the findings.

We look forward to receiving your revised manuscript.

Kind regards,

Lorenzo Righi

Academic Editor

PLOS ONE

Journal Requirements:

"This study was supported by intramural research funding for general professor in Yonsei University College of Medicine (6-2021-0224)."

Reviewers' comments:

Reviewer's Responses to Questions

**Comments to the Author**

1. Is the manuscript technically sound, and do the data support the conclusions?

Reviewer #1: Yes

Reviewer #2: Partly

2. Has the statistical analysis been performed appropriately and rigorously? 

Reviewer #1: Yes

Reviewer #2: No

3. Have the authors made all data underlying the findings in their manuscript fully available?

Reviewer #1: Yes

Reviewer #2: No

4. Is the manuscript presented in an intelligible fashion and written in standard English?

Reviewer #1: Yes

Reviewer #2: Yes

5. Review Comments to the Author

Reviewer #1: Greetings, based on my review, the document complies with the requirements for publication. However, it is recommended to specify the study approach, whether it is qualitative or quantitative, the type of design (experimental or non-experimental), and the sampling method (probabilistic or non-probabilistic: simple random, convenience, etc.). Additionally, it is important to specify the selection criteria according to the study and the reasons behind this selection.

Reviewer #2: Thank you for the opportunity to review, here are my comments:

1) p. 4 line 54 should be modified "can potentially cause stress and eventual burnout"

2) p. 3 lines 53-63: Just reading this should show any reader that there is a major problem in how burnout prevalence is assessed. 17.6% to 76% is a ridiculous range.

Rotenstein, L. S., Torre, M., Ramos, M. A., Rosales, R. C., Guille, C., Sen, S., & Mata, D. A. (2018). Prevalence of burnout among physicians: a systematic review. Jama, 320(11), 1131-1150.

3) p. 4 line 83: Are there only 303 hospitalists in Korea? Or was this the sample you ended up with after inviting all hospitalists to participate?

4) Provide a correlation table with burnout, burnout components, DASS components and Insomina index and comment on the overlaps present between the variables in your study.

5) Exhaustion is the main component of burnout, and burnout is a syndrome, so why focus only on depersonalization and personal accomplishment. Comment on the exhaustion levels.

Canu, I. G., Marca, S. C., Dell’Oro, F., Balázs, Á., Bergamaschi, E., Besse, C., ... & Wahlen, A. (2021). Harmonized definition of occupational burnout: A systematic review, semantic analysis, and Delphi consensus in 29 countries. Scandinavian journal of work, environment & health, 47(2), 95.

Schaufeli, W. (2021). The burnout enigma solved?. Scandinavian journal of work, environment & health, 47(3), 169.

6) Again, in your discussion you say that your burnout prevalence is approximately 50%. This is not accurate, burnout cannot be diagnosed. You are only estimating risk. So if anything you are estimating burnout risk prevalence, not burnout itself. This should be emphasized. Again, I am somewhat taken aback that exhaustion is not considered here.

Maslach, C., & Leiter, M. P. (2021). How to measure burnout accurately and ethically. Harvard Business Review, 7. https://hbr.org/2021/03/how-to-measure-burnout-accurately-and-ethically

7) It would also be interesting to see a cross-tabulation of your burnout risk categories with the DASS categories.

6. PLOS authors have the option to publish the peer review history of their article (what does this mean? ). If published, this will include your full peer review and any attached files.

**Do you want your identity to be public for this peer review?** For information about this choice, including consent withdrawal, please see our Privacy Policy .

Reviewer #1: No

Reviewer #2: No

---

## [Author Response · Author response to Decision Letter 1]

14 Dec 2024

Reviewer #1: Greetings, based on my review, the document complies with the requirements for publication. However, it is recommended to specify the study approach, whether it is qualitative or quantitative, the type of design (experimental or non-experimental), and the sampling method (probabilistic or non-probabilistic: simple random, convenience, etc.). Additionally, it is important to specify the selection criteria according to the study and the reasons behind this selection.

Response: Thank you for your valuable feedback. This study is a quantitative, non-experimental study designed to assess burnout and its associated factors among hospitalists. The study included all hospitalists registered as members of the Korean Association of Internal Medicine and Korean Society of Hospital Medicine (n=303) out of 384 registered Korean hospitalists at the time of the survey. This information has been added to the Methods section for greater clarity. We revised the manuscript as follows.

<Material and Methods, Study design and target population, First paragraph>

From: The study was a cross-sectional survey conducted over 20 days, from January 30 to February 18, 2023. The target population comprised all hospitalists in Korea (n=303).

To: The study was a cross-sectional survey conducted over 20 days, from January 30 to February 18, 2023. Among 384 registered Korean hospitalists as of February 2023, the target population comprised 303 hospitalists who were members of the Korean Association of Internal Medicine and Korean Society of Hospital Medicine.

Reviewer #2: Thank you for the opportunity to review, here are my comments:

1) p. 4 line 54 should be modified "can potentially cause stress and eventual burnout"

Response: Thank you for pointing out this wording. We agree that it can be improved for clarity. The phrase has been modified to more accurately convey the progression from stress to burnout.

<Introduction, Second paragraph>

From: Despite the benefits of introducing hospitalists to hospitals, focusing solely on their productivity can potentially cause burnout and stress.

To: Despite the benefits of introducing hospitalists to hospitals, focusing solely on their productivity can potentially cause stress and eventual burnout.

2) p. 3 lines 53-63: Just reading this should show any reader that there is a major problem in how burnout prevalence is assessed. 17.6% to 76% is a ridiculous range.

Rotenstein, L. S., Torre, M., Ramos, M. A., Rosales, R. C., Guille, C., Sen, S., & Mata, D. A. (2018). Prevalence of burnout among physicians: a systematic review. Jama, 320(11), 1131-1150.

Response: Thank you for highlighting this concern regarding the wide range in reported burnout prevalence. As you noted, the range of 17.6% to 76% reflects significant variability in assessment methods, measurement tools, study populations, and burnout definitions across studies. We have revised this section to provide better context and added a citation to Rotenstein et al. (2018) to underscore the issue of inconsistent burnout assessment methods in existing literature.

<Introduction, Second paragraph>

From: Medical professionals are vulnerable to burnout. According to a review of burnout among medical residents, the burnout rate varied from 17.6% to 76.0%. [9] Another study claimed that burnout among doctors ranged from 12% to 61%. [10]

To: "Medical professionals are vulnerable to burnout. Although interpretation requires caution due to differences in assessment methods, measurement tools, and study populations [9], according to a review of burnout among medical residents, the burnout rate varied from 17.6% to 76.0%. [10] Another study claimed that burnout among doctors ranged from 12% to 61%. [11]"

Ref.

[9] Rotenstein LS, Torre M, Ramos MA, et al. Prevalence of Burnout Among Physicians: A Systematic Review. JAMA. 2018;320(11):1131-1150. doi:10.1001/jama.2018.12777

3) p. 4 line 83: Are there only 303 hospitalists in Korea? Or was this the sample you ended up with after inviting all hospitalists to participate?

Response: Thank you for seeking clarification. At the time of the study, there were 384 hospitalists actively working in Korea. The study included all hospitalists registered as members of the Korean Association of Internal Medicine and Korean Society of Hospital Medicine (n=303) out of 384 registered Korean hospitalists at the time of the survey. We sent an online survey link to these members through e-mail and text messages. We have clarified this in the manuscript to avoid any ambiguity.

<Material and Methods, Study design and target population, First paragraph>

From: The study was a cross-sectional survey conducted over 20 days, from January 30 to February 18, 2023. The target population comprised all hospitalists in Korea (n=303). The online survey link was sent to the participants through e-mail and text messages by the Korean Association of Internal Medicine and Korean Society of Hospital Medicine.

To: The study was a cross-sectional survey conducted over 20 days, from January 30 to February 18, 2023. Among 384 registered Korean hospitalists, the target population comprised 303 hospitalists who were members of the Korean Association of Internal Medicine and Korean Society of Hospital Medicine. The online survey link was sent through e-mail and text messages.

4) Provide a correlation table with burnout, burnout components, DASS components and Insomnia index and comment on the overlaps present between the variables in your study.

Response: We present a correlation table between the total burnout score and its components, the total DASS score and its components, and the Insomnia Index.

The correlation analysis revealed statistically significant relationships between the burnout total score and burnout components, DASS components, and the Insomnia Index, with the exception of Reduced Personal Accomplishment and ISI. Among the DASS components—Depression, Anxiety, and Stress—the correlation was strongest between the Burnout total score and Depression (0.558, p<0.01), followed by Stress (0.539, p<0.01) and Anxiety (0.477, p<0.01). When examining individual burnout components, we found that Emotional Exhaustion, Depersonalization, and Reduced Personal Accomplishment all showed the highest correlation with Depression. For the Insomnia Index, the highest correlation was observed with Emotional Exhaustion.

<Result, Burnout and psychiatric symptoms among hospitalist, First paragraph>

From: More than half of the respondents screened positively for two burnout domains: 50.6% (n=40) for depersonalization, 57.0% (n=45) for reduced personal accomplishment, and 10.1% (n=8) reported high degree of emotional exhaustion (Figure 1). Of the respondents, 7.5% (n=6) reported mild to moderate depressive symptoms and 5.1% (n=4) reported mild to moderate anxiety symptoms. None of the respondents screened positive for severe depression, anxiety, or stress. Approximately 30.4% (n=24) of the respondents reported experiencing subthreshold insomnia and 8.9% (n=7) were screened for moderate to severe insomnia (Table 2).

To: More than half of the respondents screened positively for two burnout domains: 50.6% (n=40) for depersonalization, 57.0% (n=45) for reduced personal accomplishment, and 10.1% (n=8) reported high degree of emotional exhaustion (Figure 1). Of the respondents, 7.5% (n=6) reported mild to moderate depressive symptoms and 5.1% (n=4) reported mild to moderate anxiety symptoms. None of the respondents screened positive for severe depression, anxiety, or stress. Approximately 30.4% (n=24) of the respondents reported experiencing subthreshold insomnia and 8.9% (n=7) were screened for moderate to severe insomnia (Table 2). An analysis of the correlations between each burnout component, DASS, and ISI revealed statistically significant correlations, with the exception of the correlation between Reduced Personal Accomplishment and ISI (Supplemental Table 1).

5) Exhaustion is the main component of burnout, and burnout is a syndrome, so why focus only on depersonalization and personal accomplishment. Comment on the exhaustion levels.

Canu, I. G., Marca, S. C., Dell’Oro, F., Balázs, Á., Bergamaschi, E., Besse, C., ... & Wahlen, A. (2021). Harmonized definition of occupational burnout: A systematic review, semantic analysis, and Delphi consensus in 29 countries. Scandinavian journal of work, environment & health, 47(2), 95.

Schaufeli, W. (2021). The burnout enigma solved?. Scandinavian journal of work, environment & health, 47(3), 169.

Response: Thank you for your suggestion. It was not our intention to exclude the emotional exhaustion domain. However, in our study, the severity of the other two domains (depersonalization and reduced personal accomplishment) were measured to be higher than that of emotional exhaustion, and in emphasizing this aspect, our wording may have led to some misunderstanding. We have revised this section as follows.

<Result, Burnout and psychiatric symptoms among hospitalist, First paragraph>

From: More than half of the respondents screened positively for two burnout domains: depersonalization and reduced personal accomplishment (Figure 1).

To: More than half of the respondents screened positively for two burnout domains: 50.6% (n=40) for depersonalization, 57.0% (n=45) for reduced personal accomplishment, and 10.1% (n=8) reported high degree of emotional exhaustion (Figure 1).

6) Again, in your discussion you say that your burnout prevalence is approximately 50%. This is not accurate, burnout cannot be diagnosed. You are only estimating risk. So if anything you are estimating burnout risk prevalence, not burnout itself. This should be emphasized. Again, I am somewhat taken aback that exhaustion is not considered here.

Maslach, C., & Leiter, M. P. (2021). How to measure burnout accurately and ethically. Harvard Business Review, 7. https://hbr.org/2021/03/how-to-measure-burnout-accurately-and-ethically

Response: Thank you for your considerate comment. As you pointed out, since we used the MBI-HSS to measure evaluate burnout, it would be more appropriate to consider it as assessing risk. In addition, we added comment about the result of emotional exhaustion of this study. The revised manuscript is as follows.

<Discussion, Second paragraph>

From: The prevalence of burnout among hospitalists in our study (approximately 50%) closely aligns with that found in previous research (30% to 60%), [11-13] including the findings in Roberts et al. (52.3% among 130 internal-medicine hospitalists). [13] The difference from previous studies lies in the fact that, while they focused on internal medicine hospitalists, we targeted hospitalists with diverse specialties. Despite this distinction, the similarity in the findings suggests a substantial burnout rate among hospitalists, irrespective of their specialization. In a previous study, high levels of emotional exhaustion and depersonalization were reported, [13] whereas in the present study, elevated levels of depersonalization and reduced personal accomplishment were observed. Emotional exhaustion tends to be strongly associated with job demand variables, including work overload, whereas depersonalization and personal accomplishment are more associated with resources, including social support or the quality of job contents. [25] This indicates that burnout among Korean hospitalists may be influenced more by the quality of their work than by the quantity of their workload.

To: The signs of burnout among hospitalists in our study, which could be lead to higher risk for burnout in Korean hospitalists, closely aligns with that found in previous research (30% to 60%), [11-13] including the findings in Roberts et al. (52.3% among 130 internal-medicine hospitalists). [13] The difference from previous studies lies in the fact that, while they focused on internal medicine hospitalists, we targeted hospitalists with diverse specialties. Despite this distinction, the similarity in the findings suggests a substantial burnout rate among hospitalists, irrespective of their specialization. In a previous study, high levels of emotional exhaustion and depersonalization were reported, [13] whereas in the present study, elevated levels of depersonalization and reduced personal accomplishment with relatively low levels of emotional exhaustion were observed. Depersonalization and personal accomplishment are more associated with resources, including social support or the quality of job contents, while emotional exhaustion tends to be strongly associated with job demand variables, including work overload. [25] Thus, the result of this study indicates that burnout among Korean hospitalists may be influenced more by the quality of their work than by the quantity of their workload.

7) It would also be interesting to see a cross-tabulation of your burnout risk categories with the DASS categories.

Response: Thank you for highlighting points that add interest to this study. By comparing burnout risk categories with DASS categories, we can create a cross-tabulation table that yields the following results.

Overall, when the burnout component was in the 'High' group, a higher proportion of participants fell into the Mild or Moderate categories of the DASS components. For Emotional Exhaustion in the 'High' group, 12.5% were in both the Mild and Moderate Depression groups, and 37.5% were in the Mild Anxiety group. For Depersonalization in the 'High' group, 10% were in the Mild Depression group, 2.5% in the Moderate Depression group, 7.5% in the Mild Anxiety group, and 2.5% in the Moderate Anxiety group. Finally, for Reduced Personal Accomplishment in the 'High' group, 11.1% were in the Mild Depression group, 2.2% in the Moderate Depression group, 6.7% in the Mild Anxiety group, and 2.2% in the Moderate Anxiety group. No participants were classified in the Mild or Moderate groups for the Stress component of the DASS.

Statistical analysis revealed no significant differences overall between burnout components and each DASS component; however, a statistically significant difference was observed between Emotional Exhaustion and Anxiety (p-value = 0.002).

<Results, Burnout and psychiatric symptoms among hospitalists, First paragraph>

From: Approximately 30.4% (n=24) of the respondents reported experiencing subthreshold insomnia and 8.9% (n=7) were screened for moderate to severe insomnia (Table 2). An analysis of the correlations between each burnout component, DASS, and ISI revealed statistically significant correlations, with the exception of the correlation between Reduced Personal Accomplishment and ISI. (Supplemental Table 1)

To: Approximately 30.4% (n=24) of the respondents reported experiencing subthreshold insomnia and 8.9% (n=7) were screened for moderate to severe insomnia (Table 2). An analysis of the correlations between each burnout component, DASS, and ISI revealed statistically significant correlations, with the exception of the correlation between Reduced Personal Accomplishment and ISI. (Supplemental Table 1) Overall, when the burnout component was in the 'High' group, the proportion of participants in the Mild or Moderate categories of the DASS components was relatively higher (Supplemental Table 2).

6. PLOS authors have the option to publish the peer review history of their article (what does this mean?). If published, this will include your full peer review and any attached files.

Other comments:

- In Table 1, it is unclear what the p-values refer to, as no comparisons are explicitly mentioned. Additionally, it would be helpful to compare the characteristics of respondents with those of the full sample of 303 hospitalists to whom the questionnaire was sent. This comparison could provide insight into any potential response bias and the representativeness of the study sample.

Response: Thank you for providing comments that help in understanding the study. As mentioned, the p-values in Table 1 were excluded due to their potential to cause confusion. These p-values were derived from statistical analyses where the burnout total score was the dependent variable, and each variable listed in Table 1: Characteristics was an independent variable. However, as these values appeared to be information

---

## [Decision Letter · Decision Letter 1]

7 Jan 2025

PONE-D-24-45133R1Prevalence, Risk and Protective Factors of Burnout among Korean HospitalistsPLOS ONE

Dear Dr. Park,

Thank you for submitting your manuscript to PLOS ONE. After careful consideration, we feel that it has merit but does not fully meet PLOS ONE’s publication criteria as it currently stands. Therefore, we invite you to submit a revised version of the manuscript that addresses the points raised during the review process.

We ask that authors consider both of the reviewers' comments/suggestions as these may affect the manuscript's conclusion. Thank you.

We look forward to receiving your revised manuscript.

Kind regards,

Che Matthew Harris

Academic Editor

PLOS ONE

Journal Requirements:

Additional Editor Comments (if provided):

Reviewers' comments:

Reviewer's Responses to Questions

**Comments to the Author**

1. If the authors have adequately addressed your comments raised in a previous round of review and you feel that this manuscript is now acceptable for publication, you may indicate that here to bypass the “Comments to the Author” section, enter your conflict of interest statement in the “Confidential to Editor” section, and submit your "Accept" recommendation.

Reviewer #2: All comments have been addressed

Reviewer #3: (No Response)

2. Is the manuscript technically sound, and do the data support the conclusions?

Reviewer #2: Yes

Reviewer #3: Yes

3. Has the statistical analysis been performed appropriately and rigorously? 

Reviewer #2: Yes

Reviewer #3: Yes

4. Have the authors made all data underlying the findings in their manuscript fully available?

Reviewer #2: Yes

Reviewer #3: Yes

5. Is the manuscript presented in an intelligible fashion and written in standard English?

Reviewer #2: Yes

Reviewer #3: Yes

6. Review Comments to the Author

Reviewer #2: I am happy that the authors made sufficient effort to address my comments. I still think the prevalence rate is a mistake and contributes to confusion on the topic - but they have explained this limitation.

The authors should however clearly state that the correlations between the components of burnout and DASS-depression were large, but well below cut-offs for discriminant validity issues. This adds to the literature.

Reviewer #3: Thank you for the opportunity to review this interesting work. I was added as a reviewer after the first revisions, so I am reviewing the revised manuscript and appreciate the work of the authors and the reviewers in the improvements that are already evident. In summary, I find the revised manuscript appropriate for publication, and my comments are more on the order of suggestions and thoughts than requests for further revision:

-Beginning in the abstract but continuing throughout, I found the phrase "maximum number of inpatients" difficult to parse. I suspect it refers to a cap on the number of patients a hospitalist could manage in a day, but it almost reads as if burnout were decreased if the hospitalist managed the maximum number of patients possible. Some sort of clarification or rewording would be helpful to the reader.

-In the Conclusions section of the abstract, the phrase "depersonalization and reduced personal accomplishments domain" seems like it should be "depersonalization and reduced personal accomplishment domains," at least if I'm understanding it correctly.

-At least on the version I received, the abstract ends abruptly (and without punctuation), suggesting at least one missing word.

-It is an unfortunate consequence of the opposite orientation of the PA component of the MBI (relative to the EE and DP components) that discussing its results can be confusing. On page 8, the wording makes it sound like having a research mentor actually lowered the subjects' sense of personal accomplishment, when the Discussion makes clear that the opposite (the expected result) is true.

-The sentence at the bottom of page 10 (carrying over to page 11) does not make sense and appears to be missing a clause. It sets up a "contrary to" that never resolves, at least in my copy.

-I understand the intended meaning, but on page 11 I would avoid the phrase "negative marital status" to refer to single physicians, using "single" or "unmarried" instead. I work with a number of unmarried physicians who consider it quite a positive status!

-In a bigger picture sense, I'm fascinated by the aspect of this data that it reflects the experiences of a country introducing hospitalists long after many other countries did so. Although the history of this development is nicely laid out in the Introduction, I feel a paragraph in the Discussion about the usefulness of this data to other countries considering using the hospitalist model could make the data even more compelling than the novelty of the first report of Korean hospitalist burnout already does.

7. PLOS authors have the option to publish the peer review history of their article (what does this mean? ). If published, this will include your full peer review and any attached files.

**Do you want your identity to be public for this peer review?** For information about this choice, including consent withdrawal, please see our Privacy Policy .

Reviewer #2: No

Reviewer #3: **Yes: ** Daniel L. Roberts, M.D.

---

## [Author Response · Author response to Decision Letter 2]

12 Feb 2025

Reviewer #2: I am happy that the authors made sufficient effort to address my comments. I still think the prevalence rate is a mistake and contributes to confusion on the topic - but they have explained this limitation.

The authors should however clearly state that the correlations between the components of burnout and DASS-depression were large, but well below cut-offs for discriminant validity issues. This adds to the literature.

Thank you for clearly identifying important aspects that enhance the completeness of the study. According to previous research, if the correlation coefficient between different scales is less than 0.8, it can be judged that there is no issue with discriminant validity. As shown in Supplemental Table 1 of this study, the absolute values of the correlation coefficients between the components of Burnout and DASS-Depression are all below 0.6, confirming that there is no issue with discriminant validity. This information has also been incorporated into the manuscript.

From: More than half of the respondents screened positively for two burnout domains: 50.6% (n=40) for depersonalization, 57.0% (n=45) for reduced personal accomplishment, and 10.1% (n=8) reported high degree of emotional exhaustion (Figure 1). Of the respondents, 7.5% (n=6) reported mild to moderate depressive symptoms and 5.1% (n=4) reported mild to moderate anxiety symptoms. None of the respondents screened positive for severe depression, anxiety, or stress. Approximately 30.4% (n=24) of the respondents reported experiencing subthreshold insomnia and 8.9% (n=7) were screened for moderate to severe insomnia (Table 2). An analysis of the correlations between each burnout component, DASS, and ISI revealed statistically significant correlations, with the exception of the correlation between Reduced Personal Accomplishment and ISI (Supplemental Table 1).

To: More than half of the respondents screened positively for two burnout domains: 50.6% (n=40) for depersonalization, 57.0% (n=45) for reduced personal accomplishment, and 10.1% (n=8) reported high degree of emotional exhaustion (Figure 1). Of the respondents, 7.5% (n=6) reported mild to moderate depressive symptoms and 5.1% (n=4) reported mild to moderate anxiety symptoms. None of the respondents screened positive for severe depression, anxiety, or stress. Approximately 30.4% (n=24) of the respondents reported experiencing subthreshold insomnia and 8.9% (n=7) were screened for moderate to severe insomnia (Table 2). The correlation analysis between each burnout component, DASS, and ISI demonstrated statistically significant associations, except for the correlation between Reduced Personal Accomplishment and ISI. The absolute values of the correlation coefficients between the components of Burnout and DASS-Depression were all below 0.6, indicating no concerns regarding discriminant validity (Supplemental Table 2).[26]

[26] Clark, L.A., & Watson D. Constructing validity: Basic issues in objective scale development. Psychological Assessment. 1995;7(3):309-319. doi:10.1037/1040-3590.7.3.309.

Reviewer #3: Thank you for the opportunity to review this interesting work. I was added as a reviewer after the first revisions, so I am reviewing the revised manuscript and appreciate the work of the authors and the reviewers in the improvements that are already evident. In summary, I find the revised manuscript appropriate for publication, and my comments are more on the order of suggestions and thoughts than requests for further revision:

-Beginning in the abstract but continuing throughout, I found the phrase "maximum number of inpatients" difficult to parse. I suspect it refers to a cap on the number of patients a hospitalist could manage in a day, but it almost reads as if burnout were decreased if the hospitalist managed the maximum number of patients possible. Some sort of clarification or rewording would be helpful to the reader.

Thank you for your comments. As you mentioned, the term "maximum number of inpatients" does seem potentially confusing. Our intention was that a higher number of inpatients was associated with reduced burnout of hospitalists. As you pointed out, this might be not an intuitive finding. Our interpretation of these results is as follows; significant association between maximum number of inpatients and reduced burnout was only observed with reduced personal accomplishment subdomain, not in depersonalization or emotional exhaustion. This finding might suggest that when the inpatient load per hospitalist is too low, it might negatively impact their sense of professional satisfaction. This is an area that warrants further exploration in future research. We have addressed this point in the discussion as follows: "Additionally, a higher number of inpatients load per hospitalist was correlated with reduced burnout, specifically contributing to an increase in personal accomplishment within burnout domains. Considering the relatively short history of the domestic hospitalist system, it can be interpreted that achieving personal accommodation is facilitated by direct patient care, autonomous treatment, and appropriate workload allocation. However, it is worth noting that contrary to the general expectation that higher workload, which might relate to higher patient volumes, could worsen burnout. Further research is needed to explore this discrepancy." To minimize confusion, we will revise the term "maximum number of inpatients" to "cap on daily inpatient load per hospitalist."

From: “More than half of the respondents reported high graded burnout for two domains: depersonalization (50.6%) and reduced personal accomplishment (57%). Conflicts with caregivers, excessive workload, and long working hours were common risk factors for both burnout domains. The satisfaction with nonclinical work was identified as protective factors in depersonalization, and the availability of a research mentor and maximum number of inpatients were protective factors in reduced personal accomplishment. In the correlation analysis, the maximum number of inpatients and hospitalists per hospital was a significant factor in reducing burnout.”

To: “More than half of the respondents reported high graded burnout for two domains: depersonalization (50.6%) and reduced personal accomplishment (57%). Conflicts with caregivers, excessive workload, and long working hours were common risk factors for both burnout domains. The satisfaction with nonclinical work was identified as protective factors in depersonalization, and the availability of a research mentor and cap on daily inpatient load per hospitalist were protective factors in reduced personal accomplishment. In the correlation analysis, the maximum number of inpatients and hospitalists per hospital was a significant factor in reducing burnout.”

* All instances of "maximum number of inpatients" throughout the manuscript have been replaced with "cap on daily inpatient load per hospitalist."

-In the Conclusions section of the abstract, the phrase “depersonalization and reduced personal accomplishments domain” seems like it should be “depersonalization and reduced personal accomplishment domains,” at least if I’m understanding it correctly.

We agree with the reviewer's comment. The phrase should indeed be corrected to "depersonalization and reduced personal accomplishment domains" since it refers to two distinct domains of burnout. We will make this grammatical correction in the revised manuscript.

-At least on the version I received, the abstract ends abruptly (and without punctuation), suggesting at least one missing word.

We appreciate the reviewer pointing this out. Indeed, the abstract ended abruptly due to the omission of "quality of medical care" in the final sentence. We have now revised the abstract to include this missing phrase and added appropriate punctuation to ensure a complete and proper conclusion.

“Conclusions: This study revealed a high graded burnout rate of more than 50% in depersonalization and reduced personal accomplishments domain among Korean hospitalists, and found the risk and protective factors against burnout. The development of targeted interventions to mitigate burnout based on this study could enhance the mental well-being of healthcare professionals and improve the overall quality of medical care.”

-It is an unfortunate consequence of the opposite orientation of the PA component of the MBI (relative to the EE and DP components) that discussing its results can be confusing. On page 8, the wording makes it sound like having a research mentor actually lowered the subjects' sense of personal accomplishment, when the Discussion makes clear that the opposite (the expected result) is true.

We agree with the reviewer's point about the potential confusion in interpreting the Personal Accomplishment (PA) component of the MBI, which is scored in the opposite direction from the Emotional Exhaustion (EE) and Depersonalization (DP) components. We have revised the text to clarify that research mentorship is actually associated with a higher sense of personal accomplishment.

From: “The maximum number of inpatients and the availability of research mentors were significantly related to low total burnout scores and reduced personal accomplishment, while satisfaction with non-clinical work was significantly related to low total burnout scores and low depersonalization scores (Table 3).”

To: “The maximum number of inpatients and the availability of research mentors were significantly associated with low total burnout scores and higher sense of personal accomplishment, while satisfaction with non-clinical work was significantly related to low total burnout scores and low depersonalization scores (Table 3).”

-The sentence at the bottom of page 10 (carrying over to page 11) does not make sense and appears to be missing a clause. It sets up a "contrary to" that never resolves, at least in my copy.

We thank the reviewer for pointing out the incomplete sentence structure. We have revised the text to complete the "contrary to" comparison by adding the resolution clause. The revised version now properly presents the contrast between the general expectation about workload and burnout, and our study findings.

From: “Considering the relatively short history of the domestic hospitalist system, it can be interpreted that achieving personal accommodation is facilitated by direct patient care, autonomous treatment, and appropriate workload allocation. However, it is worth noting that contrary to the general expectation that higher workload, which might relate to higher patient volumes, could worsen burnout. Further research is needed to explore this discrepancy.”

To: "Considering the relatively short history of the domestic hospitalist system, it can be interpreted that achieving personal accommodation is facilitated by direct patient care, autonomous treatment, and appropriate workload allocation. Contrary to the general expectation that higher workload and higher patient volumes could worsen burnout, our findings suggest a different pattern. Further research is needed to explore this discrepancy."

-I understand the intended meaning, but on page 11 I would avoid the phrase "negative marital status" to refer to single physicians, using "single" or "unmarried" instead. I work with a number of unmarried physicians who consider it quite a positive status!

We appreciate the reviewer's thoughtful feedback regarding the terminology. We agree that the phrase "negative marital status" could imply an unintended value judgment. We have revised the text to use "unmarried" instead, which is a more neutral and appropriate term to describe marital status.

From: “Certain demographic factors, including young age, female sex, and negative marital status are also risk factors for vulnerability to burnout.”

To: “Certain demographic factors, including young age, female sex, and unmarried are also risk factors for vulnerability to burnout.”

-In a bigger picture sense, I'm fascinated by the aspect of this data that it reflects the experiences of a country introducing hospitalists long after many other countries did so. Although the history of this development is nicely laid out in the Introduction, I feel a paragraph in the Discussion about the usefulness of this data to other countries considering using the hospitalist model could make the data even more compelling than the novelty of the first report of Korean hospitalist burnout already does.

Thank you for your novel suggestion. We added about the usefulness of our data to other countries using the hospitalist models in the discussion.

From: “Nonetheless, the significance of this study stems from its distinction as the first to examine burnout and psychiatric symptoms among hospitalists.”

To: “Nonetheless, the significance of this study stems from its distinction as the first to examine burnout and psychiatric symptoms among hospitalists during the early adoption phase of the hospitalist system in South Korea. Unlike the United States, where the system has been established for over 20 years and is relatively well-defined, this study explores the experience of South Korea, where the hospitalist system is still in its early stages of implementation. The findings of this study hold significance as they may serve as a valuable reference for other countries that are considering implementing the hospitalist system or are in the initial phase of its adoption.”

---

## [Editor Report · Decision Letter 2]

14 Feb 2025

Prevalence, Risk and Protective Factors of Burnout among Korean Hospitalists

PONE-D-24-45133R2

Dear Dr. Park,

We’re pleased to inform you that your manuscript has been judged scientifically suitable for publication and will be formally accepted for publication once it meets all outstanding technical requirements.

Kind regards,

Che Matthew Harris

Academic Editor

PLOS ONE
---

## [Editor Report · Acceptance letter]

PONE-D-24-45133R2

PLOS ONE

Dear Dr. Park,

I'm pleased to inform you that your manuscript has been deemed suitable for publication in PLOS ONE. Congratulations! Your manuscript is now being handed over to our production team.

Kind regards,

on behalf of

Dr. Che Matthew Harris

Academic Editor

PLOS ONE